

# Characterization of the populations of upside-down jellyfish in Jardines de la Reina National Park, Cuba

Ramón Damián Morejón-Arrojo[1,2] and Leandro Rodriguez-Viera[1]

[1] Center for Marine Research, University of Havana, La Habana, Cuba
[2] Faculty of Biology, Univerity of Havana, La Habana, Cuba

## ABSTRACT

Upside-down jellyfish are a group of benthic scyphozoans belonging to the genus *Cassiopea*, whose members are in symbiosis with dinoflagellates and inhabit tropical and subtropical waters. Although there are some studies of the genus in the Caribbean, these are limited. In Cuba, the group's studies are restricted to reports on taxonomic lists and, as far as we know, no one has performed any analyzes of the densities of these jellyfish in seagrass or mangrove ecosystems in Cuba. In this work, the populations of *Cassiopea* spp. in Jardines de la Reina National Park (JRNP) were characterized, for the first time for this Marine Protected Area and Cuba. One hundred $1m^2$ square frames were placed at 14 JRNP sites. For each site, the species, density, size of the individuals and abiotic factors were determined. Density and diameter comparisons were made between sites, zones and regions within the JRNP. The percentage of the benthic cover was determined and a correlation was made between density and diameter. A total of 10,803 individuals were recorded, of which 7,618 belong to *Cassiopea xamachana* and 3,185 belong to *Cassiopea frondosa*. Both species share a niche and no evident segregation was detected according to abiotic variables. Significant differences were detected in comparisons of density and size across sites and zones. Density and size in the JRNP were negatively correlated, and higher aggregations of the species were observed at lower sizes. Density mean values ranged from 2.18 to 14.52 ind. /$m^2$ with maximum values of 79 ind. /$m^2$. Cayo Alcatraz was the site found to have the highest density while Cachiboca was the site with the lowest density. The average bell diameter size of the individuals ranged from 9.34 to 15.31 cm for the sampled sites, with minimum and maximum values of 2.5 cm and 32.6 cm. The smallest size was recorded at Cayo Alcatraz while the largest size was reported for Boca de las Anclitas. The environmental factors evaluated showed no significant relationship with the density or diameter of *Cassiopea*, while the *Thalassia testudinum* cover was negatively correlated with *Cassiopea* density at all fourteen sites in the JRNP. The percentage of *Cassiopea* coverage was higher than those reported in the literature, with four sites exceeding 20% coverage. In general, the populations of *Cassiopea* spp. in the JRNP did not differ greatly, although a higher density was observed towards the eastern region of the park. It was shown for the first time for the species that density and size have a negative correlation. Future studies are required to quantify the impact of *Cassiopea* on coastal marine ecosystem processes, and to further determine how anthropogenic changes may be altering the function of these tropical ecosystems.

Corresponding author
Leandro Rodriguez-Viera,
leokarma@gmail.com

## INTRODUCTION

The genus *Cassiopea* Peron and Lesueur 1810 consists of twelve species, known as the upside-down jellyfish, a benthic scyphozoan (Rhizostomeae), which is found in symbiosis with dinoflagellates of the genus *Symbiodinium* (*Lampert, 2016*). Unlike other jellyfish, these coelenterates exhibit an epibenthic lifestyle with their inverted bell on the substrate and upward-facing oral arms, which is an adaptation to symbiosis with dinoflagellates (*Lampert, 2016*). These jellyfish harbor an abundant community of symbionts, which together with their lifestyle favour light uptake by symbionts (*Verde & McCloskey, 1998*). The relationship with zooxanthellae provides *Cassiopea* with some of the nutrients they need, however, they are known to feed in turn on fish larvae, copepods, larvae, and small crustaceans (*Larson, 1997*).

*Cassiopea* is a keystone organism for many habitats, feeding pelagic food webs on reefs by releasing organic matter and playing an essential role in nutrient cycling (*Jantzen et al., 2010*; *Niggl et al., 2010*). The mucus expelled by *Cassiopea* through the contraction of the bell collects the suspended particles, thus preventing sedimentation and also these pulses keep the degraded nutrients of the mangroves moving in the water column. (*Ohdera et al., 2018*; *Durieux et al., 2021*). Large populations of jellyfish directly favor the circulation of nutrients from mangroves to reefs, which allows the development of fish and corals in coral reefs (*Niggl et al., 2010*).

This genus of jellyfish has gained attention in recent years, being studied as: bioindicator species for its ability to tolerate elevated metal concentrations, with applications for coastal ecosystem management (*Klein, Pitt & Carroll, 2016*; *Klein et al., 2017*; *Templeman, McKenzie & Kingsford, 2021*), as a model for the study of cnidarian-zooxanthellae symbiosis (*Newkirk, Frazer & Martindale, 2018*; *Medina et al., 2021*), and also as model in studies of fluid dynamics (*Durieux et al., 2021*; *Battista et al., 2022*).

*Cassiopea* distributed mainly in the South Atlantic (Brazil), the Caribbean Sea, and the Indo-Pacific Sea inhabiting shallow tropical and subtropical waters such as mangroves and seagrass beds (*Morandini et al., 2017*), although their distribution has spread due to invasion into the Mediterranean Sea (*Cillari, Andoloro & Castriota, 2018*). Evidence in recent studies shows that species of *Cassiopea* are increasing their potential as invasive species by showing tolerance and adaptation to fluctuations in environmental factors such as salinity, temperature, and pH (*Morandini et al., 2017*; *Weeks et al., 2019*). In addition their asexual reproduction and tissue regeneration, combined with their feeding style, make some species apt to conquer different habitats and tolerate environmental variations (*Gamero-Mora et al., 2019*; *Mammone et al., 2021*).

Most studies on this group of jellyfish comprise different approaches: genetic (*Stampar et al., 2020*; *Ames et al., 2021*; *Gamero-Mora et al., 2022*), ecological (*Iliff et al., 2020*; *Zarnoch et al., 2020*; *Stoner, Archer & Layman, 2022*), and symbiosis with zooxanthellae (*Newkirk et al., 2020*; *Olguin-Jacobson & Pitt, 2021*). In the Caribbean region, there are several studies of the genus that address ecological aspects (*Stoner et al., 2011*; *Stoner, Yeager & Layman, 2014a*; *Stoner et al., 2014b*; *Stoner, Archer & Layman, 2022*; *Fitt et al., 2021*) and are included in some lists of jellyfishes species (*Ehemann, Riggio & Cedeño Posso,*
_2015_; _Mendoza-Becerril & Agüero, 2019_). However, ecological studies are limited. In Cuba, studies in this group are limited to records in species lists, reporting the presence of _Cassiopea xamachana Bigelow, 1892_ and _Cassiopea frondosa_ (Pallas, 1774) on the Cuban shelf (_Rodríguez-Viera et al., 2012_; _Semidey et al., 2013_). There is no ecological study of the species in Cuba today. Therefore, the objective of this research is to characterize for the first time the populations of _Cassiopea_ spp. in the Jardines de la Reina National Park (JRNP), Cuba.

## MATERIAL & METHODS

### Study area

Jardines de la Reina Archipelago extends from the Gulf of Guacanayabo to Casilda Bay, in the southern part of the island of Cuba. It has an extension of 135 kilometers and is formed by 661 keys corresponding to three big keys, of which the most extensive is Las Doce Leguas, located in the extreme west, off the southern coast of the provinces of Ciego de Avila and Camagüey (_Figueredo-Martín, Pina-Amargós & Angulo-Valdés, 2014_). Because of its ecological importance and conservation status, it was granted the category of a National Park in 2010. It is internationally recognized for the excellent state of conservation of its ecosystems (_Figueredo-Martín et al., 2010_) and constitutes one of the largest marine reserve in the Caribbean with 200,957 hectares (_Appeldoorn & Lindeman, 2003_; _Perera-Valderrama et al., 2018_). The JRNP is composed of a long chain of keys, with mangrove forests, forming estuaries with a mostly muddy substrate that presents a water current and shallow interior lagoons with seagrasses. The keys display a line of coral reefs bordering the Caribbean Sea made up of shallow escarpments (8 to 15 m) and reef crests (1 to 4 m). To facilitate the work given the size of the protected area, it was decided to divide the JRNP into three regions, which in turn are further subdivided into eight zones, numbered from 1 to 8 from east to west. The Eastern Region encompasses Zones 1, 2, and 3, the Central Region Zones 4 and 5, and the Western Region Zones 6, 7, and 8. Zone 1 is composed of the Mexicana site, Zone 2: Cayo Juan Grin and Peralta, Zone 3: Cachiboca, Zone 4: Boca de las Anclitas, Laguna de las Anclitas and Cayo Piedra Piloto, Zone 5: Canal de las Auras, Estero de las Guasas Este, Canal de Caballones and Laguna de las Anclitas Noroeste, Zone 6: Punta Oeste de Boca Grande, Zone 7: Cayo Alcatraz and Zone 8: Laguna de Bretón. The protection gradient established by _Pina-Amargós et al. (2014)_ refers to the fact that, in the center of the reserve, patrolling by inspectors and other JRNP personnel is stricter due to the proximity, while for the ends of the reserve it becomes more complex for personnel to reach the zone, which allows the incursion of illegal fishermen into the zone affecting fish populations in the area. Sampling was carried out at fourteen sites in the JRNP (Fig. 1, Table S1).

### Sampling methodology

Sampling was conducted from December 12 to 15, 2021 (Cachiboca, Canal de las Auras, Boca de las Anclitas, Laguna de las Anclitas and Peralta), from February 9 to March 3, 2022 (Cayo Piedra Piloto, Estero de las Guasas Este, Canal de Caballones, Cayo Juan Grin, Laguna de las Anclitas Noroeste and Mexicana) and from July 30 to August 4, 2022 (Cayo Alcatraz, Laguna de Bretón and Punta Oeste de Boca Grande). The field
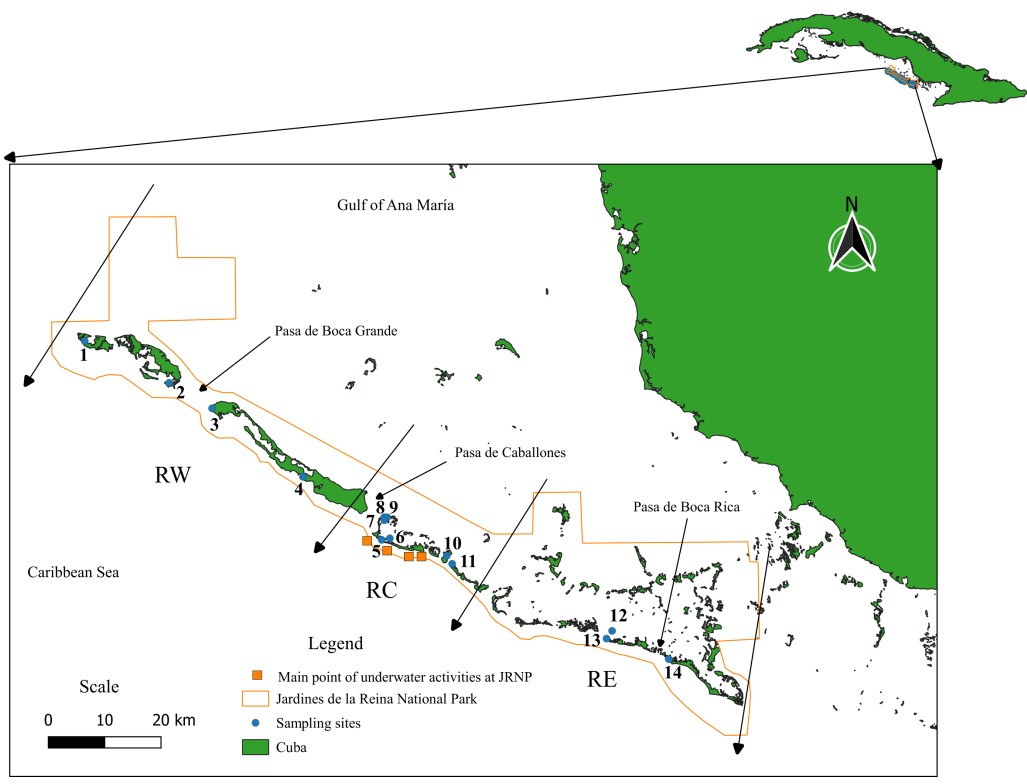

**Figure 1** **Study area. Jardines de la Reina National Park.** Sampling sites: (1) Laguna de Bretón, (2) Cayo Alcatraz, (3) Punta Oeste de Boca Grande, (4) Canal de las Auras, (5) Boca de las Anclitas, (6) Laguna de las Anclitas, (7) Laguna de las Anclitas Noroeste, (8) Estero de las Guasas Este, (9) Canal de Caballones, (10) Cayo Piedra Piloto, (11) Cachiboca, (12) Cayo Juan Grin, (13) Peralta, (14) Mexicana. Regions of Jardines de la Reina National Park taking into account the protection gradient established by *Pina-Amargós et al. (2014)*. RW, Reserve West; RC, Reserve Center; RE, Reserve East. The Eastern Region encompasses Zones 1, 2, and 3, the Central Region Zones 4 and 5, and the Western Region Zones 6, 7, and 8. The orange squares represent the main points of underwater activities carried out in the center of the Reserve (*e.g.*, diving and recreational fishing).

experiments were approved by the Oficina de Regulación de Seguridad Ambiental, No. 5 of 2021, from the Ministry of Science, Technology and Environment. For each site, the plot method was carried out, using a square frame of 1 m², which was randomly placed leaving at least five meters distance between each replicate ($N = 100$ per site). Photographs were taken in each plot using a Nikon COOLPIX Nikkor 5x optical camera. The number of *Cassiopea* spp. individuals were determined, as well as the mean size of the bell diameter (cm) for each plot using ImageJ 1.53e software, establishing the plot length (100 cm) as the scale. At the same time, physicochemical parameters were determined: pH, dissolved oxygen, temperature, and salinity, using a Hanna HI 9829 multiparameter. The two species reported (*C. frondosa* and *C. xamachana*) for Cuba were identified taking into account visible morphological characters (*e.g.*, shape, size, and color of appendages) to compare the frequency of occurrence in the fourteen sampled sites. Both species present some morphological differences that are easily distinguishable, which

makes a genetic test not necessary for their identification. *C. xamachana* has 10 to 15 oral arms on alternate branches while the oral arms of *C. frondosa* are bifurcate, with short pinnate lateral branches (*Morandini et al., 2017*; *Ohdera et al., 2018*). Finally, an important characteristic to differentiate both species is the shape of the vesicles: *C. xamachana* presents large and small ribbon-like filaments and *C. frondosa* shows leaf-shaped vesicles (*Larson, 1997*; *Ohdera et al., 2018*). In addition, the vesicles of *C. xamachana* due to the presence of zooxanthellae can be blue, black, white, yellow, or green; besides the medusae of *C. frondosa* are darker in color than those of *C. xamachana* due to the present different symbiont species/number (*Fitt et al., 2021*).

The benthic cover was estimated from the images of the square frames that were taken. Each 1 m$^2$ frame was divided into four quadrants and estimated at the rate of 25% in each quadrant taking into account the cover representation of *Cassiopea* spp., *T. testudinum*, *Syringodium filiforme*, sand (substrate), algae, and other invertebrates.

### Statistical analyses

All data were tested for normality and homogeneity of variances using the Lilliefors, Shapiro-Wills, Kolmogorov–Smirnov, and Levene tests, respectively. A comparison was made between the eight zones of the JRNP taking into account density and size. In addition, a comparison of density and size was made taking into account the regions into which the JRNP is divided (East Region, Central Region, and West Region) (Fig. 1). For both comparisons (by zones and by regions), simple Kruskal-Wallis rank-rank analysis ($p \leq 0.05$) and multiple comparisons were performed when necessary. Comparisons of diameter and density between zones and regions were made at the genus level to describe *Cassiopea* populations in JRNP. However, due to the morphological differences between the two species present in Cuba, it was possible to identify them in order to compare their frequency of occurrence.

A Spearman correlation was established, taking into account that the variables used do not meet the assumptions of normality and homogeneity of variances, to check if there is any relationship between density and mean size of *Cassiopea* spp. in the populations of the sites sampled in the JRNP. In addition, correlations were made between the variables evaluated in *Cassiopea* spp. and environmental factors.

The spatial distribution of *Cassiopea* spp. was calculated using in Morisita Index to determine if the inverted jellyfish present a random, uniform, or aggregated (patchy) distribution at each of the sites (*Morisita, 1959*; *Hayes & Castillo, 2017*). If the distribution is random, ($I\delta$) $=1$, if it is perfectly uniform ($I\delta$) $< 1.0$, and the aggregated pattern is given by ($I\delta$) $> 1.0$ with the maximum aggregation being $I\delta = n$ (when all individuals are found in one sampling unit). Statistica 10.0 (StatSoftInc., Tulsa, OK, USA) and GraphPad Prism 9.3.1.00 (GraphPad Software, Inc., San Diego, California, USA) statistical software were used for all tests and graphing.

## RESULTS

The total number of *Cassiopea* spp. individuals quantified for the fourteen sites sampled in JRNP was 11,563. Cachiboca reported the lowest number with 142 individuals, while the

**Table 1** Abundance (number of individuals) of upside-down jellyfish and Morisita index (I $\delta$) to determine the spatial distribution of *Cassiopea* spp. populations at fourteen sites in Jardines de la Reina National Park, Cuba.

| Site | C. xamachana | C. frondosa | Cassiopea spp. | I $\delta$ |
|---|---|---|---|---|
| *Cachiboca* | 100 | 42 | | 1.19 |
| *Canal de las Auras* | 223 | 273 | | 1.50 |
| *Boca de las Anclitas* | 522 | 382 | | 1.27 |
| *Laguna de las Anclitas* | 863 | 361 | | 2.10 |
| *Peralta* | 467 | 603 | 335 | 2.36 |
| *Cayo Piedra Piloto* | 109 | 41 | | 1.54 |
| *Estero de las Guasas Este* | 114 | 44 | | 1.08 |
| *Canal de Caballones* | 1896 | 470 | | 1.43 |
| *Cayo Juan Grin* | 816 | 159 | | 1.22 |
| *Laguna de las Anclitas Noroeste* | 204 | 54 | | 1.16 |
| *Mexicana* | 801 | 61 | | 1.60 |
| *Cayo Alcatraz* | 932 | 302 | 425 | 1.25 |
| *Laguna de Bretón* | 246 | 216 | | 1.47 |
| *Punta Oeste de Boca Grande* | 325 | 177 | | 1.45 |

highest number was reported by Canal de Caballones with 2,366 individuals (Table 1). For JRNP the density of *Cassiopea* was 7.98 $\pm$ 0.20 ind. /m$^2$ while the mean size was 12.53 $\pm$ 0.08 cm. The mean density values found in the fourteen sites sampled in the JRNP ranged from 2.18 $\pm$ 0.21 ind. /m$^2$ to 14.52 $\pm$ 0.89 ind. /m$^2$. The mean sizes (bell diameter) in the eight study areas ranged between 9.34 $\pm$ 0.18 cm and 15.31 $\pm$ 0.24 cm, with minimum and maximum values of 2.5 cm and 32.6 cm. The maximum value of 32.6 cm diameter belongs to an individual of *C. xamachana*, while the highest value found in this study for *C. frondosa* is 27 cm.

The depth of the sampling sites varied from 0.40- 4 m. The pH values ranged from 6.95 to 8.25. Salinity ranged from 32.25 to 37.32 PSU, while temperature ranged from 26.8 to 30 °C (Table 2).

The bell diameter analysis of *Cassiopea* spp. individuals among the eight zones into which the JRNJ is subdivided show significant differences among them (Kruskal-Wallis $H = 400.8$; $N = 1400$; $p < 0.0001$). Zones 1, 4, 6, and 8 presented the largest bell diameter sizes, with no significant differences among them. While zones 2, 3, and 5 presented mean length values (Fig. 2). Zone 7 presented the least size values with significant differences among the rest of the zones. The mean value of the bell diameter for Zone 1 was 14.71 $\pm$ 0.25 cm, for Zone 2 it was 11.31 $\pm$ 0.16 cm, for Zone 3 11.75 $\pm$ 0.40 cm, for Zone 4 14.39 $\pm$ 0.16, for Zone 5 11.39 $\pm$ 0.13 cm, Zone 6 13.38 $\pm$ 0.27 cm, for Zone 7 it was 9.33 $\pm$ 0.18 cm and 14.54 $\pm$ 0.38 cm for Zone 8 (Fig. 2).The highest mean density value was found in Zone 7 with 14.52 $\pm$ 0.89 ind. /m$^2$, while Zone 3 presented the lowest mean density value among all sites with 2.18 $\pm$ 0.21 ind. /m$^2$ (Fig. 2).

When comparing the density of the three regions of the JRNP, significant differences were found (Kruskal-Wallis, $H = 26.50$; $N = 1400$; $p < 0.0001$). The Eastern Region has the highest mean density of *Cassiopeas* in the park (9.57 $\pm$ 0.52 ind. /m$^2$, SEM), followed

**Table 2  Abiotic factors in fourteen distribution sites of upside-down jellyfish in the Jardines de la Reina National Park, Cuba.**

| Site | Depth (m) | pH | Salinity (PSU) | Dissolved oxygen (%) | Temp. (⁰C) |
|---|---|---|---|---|---|
| *Cachiboca* | 1–1.25 | 8.25 | 35.2 | 80.5 | 27.63 |
| *Canal de las Auras* | 1–1.50 | 7.9 | 35.08 | 46.5 | 27 |
| *Boca de las Anclitas* | 1–2.25 | 8 | 34.73 | 47.6 | 26.84 |
| *Laguna de las Anclitas* | 1–2 | 7.8 | 34.55 | 62.1 | 27.54 |
| *Peralta* | 0.40–2.30 | 8.14 | 34.8 | 47.5 | 28 |
| *Cayo Piedra Piloto* | 1.50 | 8.06 | 34.60 | 134.0 | 28 |
| *Estero de las Guasas Este* | 0.20–1.50 | 7.68 | 34.60 | 121.2 | 27 |
| *Canal de Caballones* | 2–4 | 7.52 | 32.25 | 120.5 | 29 |
| *Cayo Juan Grin* | 1–2.40 | 7.98 | 34.69 | 142.0 | 26 |
| *Laguna de las Anclitas Noroeste* | 0.80–1.10 | 7.49 | 34.9 | 78.5 | 27 |
| *Mexicana* | 0.50–1.50 | 8.23 | 34.20 | 132.7 | 29 |
| *Cayo Alcatraz* | 1.5–2 | 6.95 | 34.8 | 110 | 30 |
| *Laguna de Bretón* | 1.5–2 | 7.41 | 35.78 | 112.9 | 29.5 |
| *Punta Oeste de Boca Grande* | 1.5 | 7.08 | 37.32 | 110.9 | 30 |

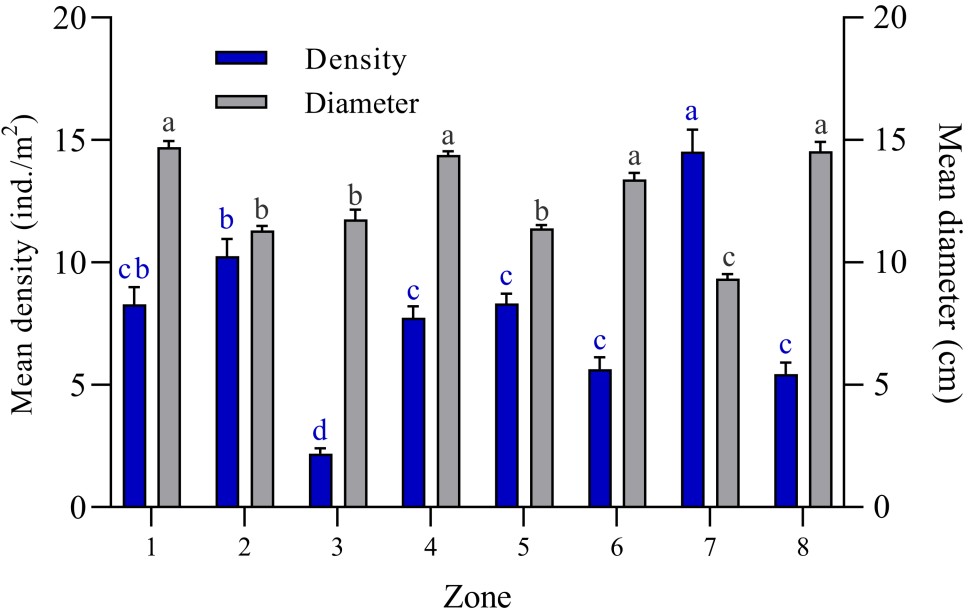

**Figure 2  Comparison of mean bell diameter (cm) and mean density (ind. /m²) of *Cassiopea* spp. in eight zones of Jardines de la Reina National Park, Cuba.** Zone 1: Mexicana, Zone 2: Peralta and Cayo Juan Grin, Zone 3: Cachiboca, Zone 4: Boca de las Anclitas, Laguna de las Anclitas and Cayo Piedra Piloto, Zone 5: Canal de las Auras, Estero de las Guasas Este, Canal de Caballones and Laguna de las Anclitas Noroeste, Zone 6: Punta Oeste de Boca Grande, Zone 7: Cayo Alcatraz and Zone 8: Laguna de Bretón. Each value is the mean ± SEM ($N = 1400$ quadrats). Differences among density are marked by different blue superscript letters (Kruskal-Wallis, $H = 400.8$; $p < 0.0001$; Tukey, $p \leq 0.05$) and differences among bell diameter size are marked by different black superscript letters (Kruskal-Wallis, $H = 197.2$; Tukey; $p \leq 0.0001$), where a represents the highest values and the lowest values.

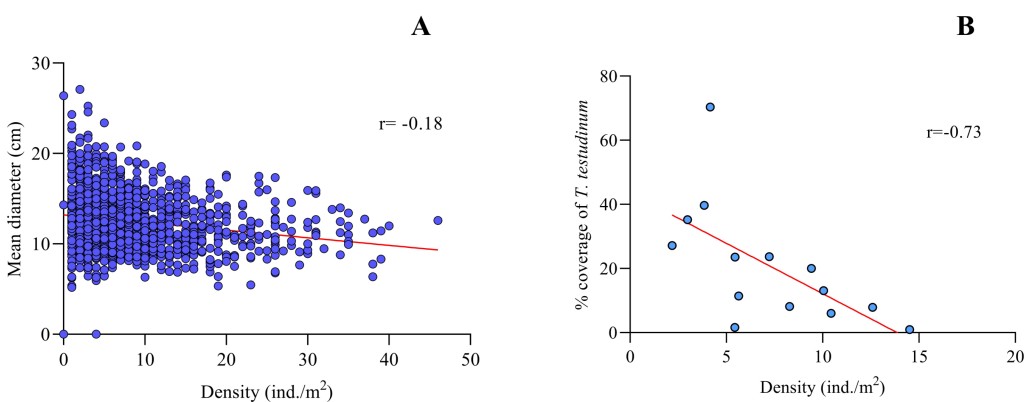

**Figure 3** **Spearman's rank correlation analysis.** (A) Density (ind/m$^2$) and mean size (cm) of *Cassiopea* spp. ($r = -0.18$; $p < 0.05$) and (B) between *Thalassia testudinum* cover (%) and density (ind. /m$^2$) of *Cassiopea* spp. ($r = -0.73$; $p < 0.05$) for fourteen sites in Jardines de la Reina National Park, Cuba.

by the Central Region (7.85 $\pm$ 0.31 ind. /m$^2$) and the Western Region (7.69 $\pm$ 0.36 ind. /m$^2$). No differences were found between the Central Region and the Western Region ($p = 0.46$), while differences were found between the Eastern Region and the Central Region ($p < 0.0001$) and the Western Region ($p = 0.0025$) (Fig. S1).

On the other hand, the size analysis between the three regions also showed significant differences (Kruskal-Wallis, $H = 6.212$; $N = 1400$; $p = 0.0448$). The largest length was found in the Central Region (12.63 $\pm$ 0.12 cm, SEM), followed by the Eastern Region (12.48 $\pm$ 0.16 cm) and finally the Western Region (12.22 $\pm$ 0.17). Significant differences were only found between the Central and Western Regions ($p = 0.0386$) (Fig. S1). No differences were found between the density and bell diameter of *Cassiopea* and the dry and rainy seasons (Mann–Whitney test, $U = 128565$, $p = 0.128$ and $U = 131624$, $p = 0.337$, respectively).

Spearman's correlation coefficient for density *vs.* mean size of *Cassiopea* spp. for the fourteen sites sampled from Jardines de la Reina evidenced a slight negative correlation between both variables ($r = -0.18$, $p < 0.0001$, $N = 1400$), with a confidence interval of 95% (Fig. 3A).

For Boca de las Anclitas and Peralta sites, the Morisita index values are 2.10 and 2.36, respectively. The rest of the sites values ranging from 1.19 to 1.60. For all the sites sampled ($I \delta$) >1.0, evidencing an aggregated distribution of *Cassiopea* spp. (Table 1).

The presence of *C. xamachana* and *C. frondosa* was recorded in all sites sampled (Fig. 4). Of the 11,563 sampled individuals, *C. xamachana* presented the highest abundance with 7,618 individuals, while 3,185 individuals were observed for *C. frondosa* (Fig. 5). Only in 760 individuals was it not possible to indicate the species level. Both species were observed at equal depths and distributed on the same type of substrate.

The percentage of *Cassiopea* benthic cover ranged from 6% at Cachiboca to 23% at Boca de las Anclitas and Laguna de las Anclitas. Of the fourteen sites, only four exceeded 20% *Cassiopea* benthic cover, including Boca de las Anclitas and Laguna de las Anclitas: Cayo

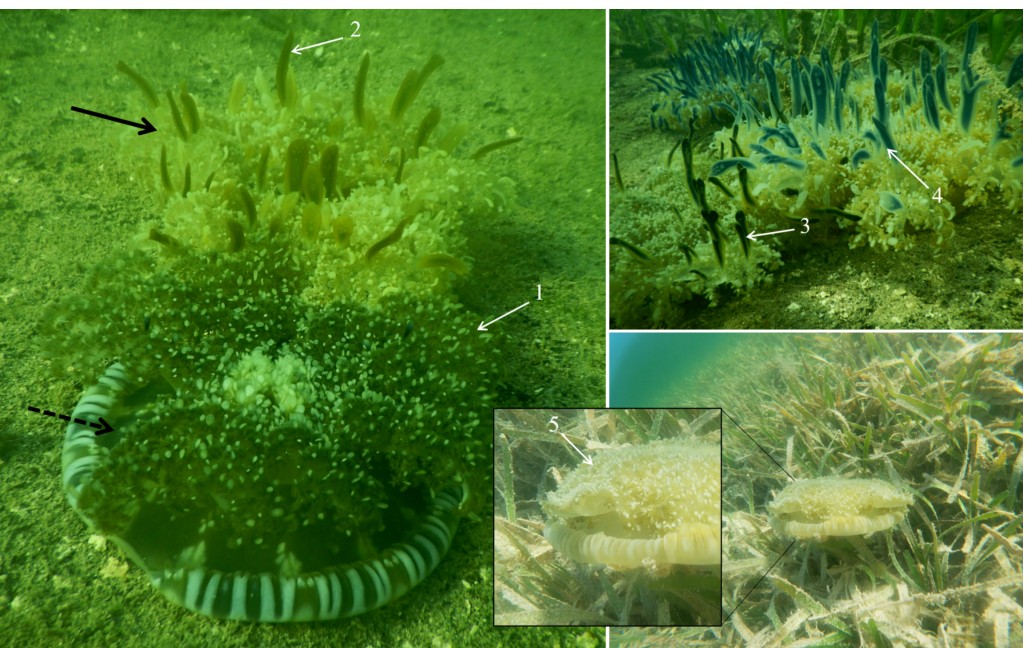

**Figure 4** **Jellyfish of the genus *Cassiopea* in mangrove canals in Jardines de la Reina National Park, Cuba.** Black arrow: *C. xamachana*, dashed black arrow: *C. frondosa*. 1 and 5. Leaf-shaped vesicles of *C. frondosa*, 2. Ribbon-like vesicles of *C. xamachana* yellow, 3. Ribbon-like vesicles of *C. xamachana* black, 4. Ribbon-like vesicles of *C. xamachana* blue in color.

Juan Grin with 20% cover and Canal de Caballones with 22% cover respectively (Fig. 6). Marine vegetation cover values at the fourteen sites were *T. testudinum*, which ranged from 0.94% at Cayo Alcatraz to 70% at Laguna de las Anclitas Noroeste. In addition, *S. filiforme* was observed at Peralta (15%), Mexicana (0.60%), Cayo Alcatraz (12.63%), and Laguna de Bretón (0.22%). Among the other invertebrates found were the anemone *Condylactis gigantea*, which represents 0.13% of total coverage, the mollusk *Strombus giga* with 0.02%, and the corals *Porites porites* and *P. asteroides* with 0.22 and 0.02% of the total respectively, in addition to holothuroids that were observed in Punta Oeste Boca Grande which represents 0.04% of total coverage.

*Thalassia testudinum* cover was negatively correlated with *Cassiopea* density at all fourteen sites in the JRNP (r =−0.73, p < 0.01) (Fig. 3B, Tables 3 and 4). The environmental factors evaluated showed no significant relationship with density or mean diameter of *Cassiopea* (Table 4).

## DISCUSSION

The salinity present at the sampling sites ranged from 32.25 to 37.32 PSU. *Klein et al. (2017)* report these salinity values as optimal for *Cassiopea* spp. *Klein, Pitt & Carroll (2016)* in their results observed that as salinity increases *Cassiopea* spp. increases by up to 8% in diameter at values of 35 PSU and for values of 17 PSU the diameter of individuals decreased by up to 16%. Therefore, salinity is an abiotic factor that may be directly related to the

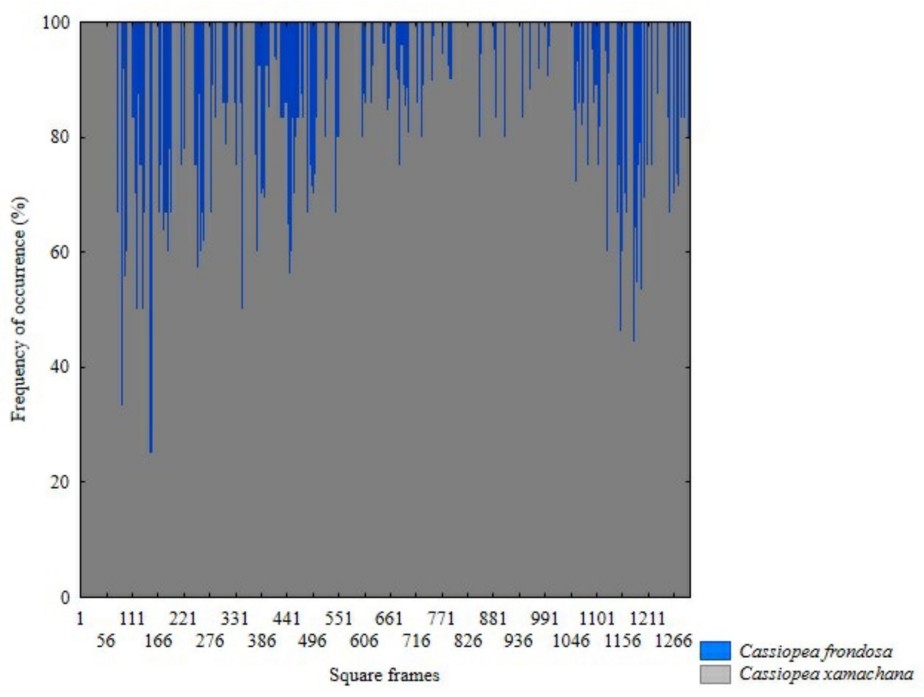

**Figure 5** Differences in frequency occurrence of *C. xamachana* and *C. frondosa* at fourteen sampled sites in Jardines de la Reina National Park, Cuba. (*N* = 1307 quadrats).

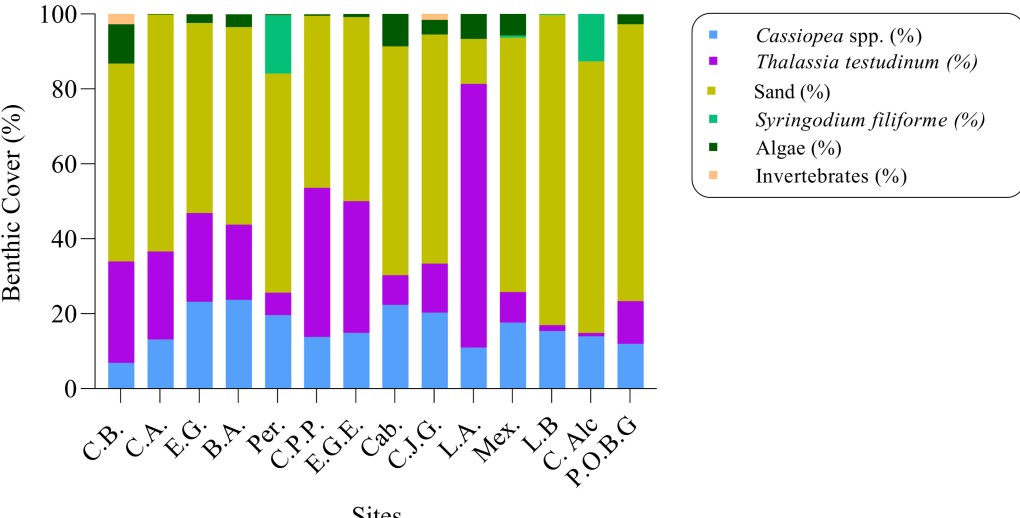

**Figure 6** Percentage of benthic cover (%) for fourteen sites in Jardines de la Reina National Park, Cuba. C.B: Cachiboca, C.A: Canal de las Auras, E.G: Estero de las Guasas, B.A: Boca de las Anclitas, Per: Peralta, C.P.P: Cayo Piedra Piloto, E.G.E: Estero de las Guasas Este, Cab: Caballones, C.J.G: Cayo Juan Grin, L.A: Laguna de las Anclitas and Mex: Mexicana.

**Table 3  Relationship between density values of *Cassiopea* spp. and the *Thalassia testudinum* coverage in the eight zones of Jardines de la Reina National Park, Cuba.** Each value is the mean ± SEM.

| Zone | Density of *Cassiopea* (ind./m²) | *Thalassia* Cover (%) |
|---|---|---|
| 1 | 8.28 ± 0.69 | 8.19 ± 1.16 |
| 2 | 10.25 ± 0.70 | 9.45 ± 1.06 |
| 3 | 2.18 ± 0.21 | 27.14 ± 2.77 |
| 4 | 7.74 ± 0.45 | 24.12 ± 1.33 |
| 5 | 8.32 ± 0.39 | 25.03 ± 1.45 |
| 6 | 5.64 ± 0.47 | 11.45 ± 1.52 |
| 7 | 14.52 ± 0.89 | 0.94 ± 0.30 |
| 8 | 5.43 ± 0.48 | 1.60 ± 0.44 |

**Table 4  Correlation values between variables determined in *Cassiopea* spp. and environmental factors (Spearman correlation, $\star$ $p < 0,05$).**

| | Mean diameter (cm) | *T. testudinum* Cover (%) | pH | Temp (°C) | Salinity (PSU) | Dissolved oxygen (%) |
|---|---|---|---|---|---|---|
| Density (ind. / m²) | −0.18* | −0.73* | −0.20 | 0.20 | −0.25 | −0.24 |
| Mean diameter (cm) | | 0.20 | 0.19 | 0.04 | −0.03 | 0.21 |

diameter of *Cassiopea* spp. individuals. A study in Barbuda by *Zarnoch et al. (2020)* reports individuals between 4.4 and 7.4 cm in bell diameter for a salinity of 38 PSU. On the other hand, in the Bahamas *Stoner et al. (2011)* obtained salinity values between 27 and 35 PSU and individuals with an average size of 12.4 cm in diameter. The salinity values during this study did not show considerable variations, therefore, salinity could not influent the differences in diameter between the sites sampled.

The temperature presented little variation (between 26.8 and 30 °C) and coincides with the optimal temperature values reported for the development and growth of these jellyfish (*Aljbour et al., 2019*). The temperature values recorded in this study moved in a narrow range, even in December, relatively warm values were recorded (26.8–28 °C), but they never exceeded 30 °C. However, the increase in temperature increases the mortality of *Cassiopea* spp. (*Newkirk, Frazer & Martindale, 2018*). *Thé et al. (2020)* recorded results showing a variation in the density of *Cassiopea* spp. according to the seasons (dry season and rainy season), finding densities between 1.32 ind. /m² and 4.55 ind. /m² during the dry season, while for the rainy season no *Cassiopea* spp. were detected. Apparently, the density of these jellyfish is usually closely related to variations in the rain season, and future studies that address this aspect in Cuba are required. The ranges of environmental parameters in this research coincide with the values reported by *Thé et al. (2020)* for the adequate growth of *Cassiopea* spp. populations. Due to some logistical problems in our study, it was not possible to measure the turbidity in all the sites, so we did not include any analysis of this abiotic factor. However, due to the characteristics of these species, it would be an important parameter to take into account in future studies.

The highest bell diameter size values among the zones of the park correspond to Zone 1, Zone 4, and Zone 8, which in turn are located in the center and at the extreme of the

JRNP. First of all, the extremes zones are the closest to land, so they are the most exposed to receiving nutrient-laden waters from the mainland. The Gulf of Ana María is characterized by the influence of the peripheral oceanic circulation, towards the northeast and especially by the formation of small cyclonic gyres (*Arriaza et al., 2008*). These cyclonic gyres may be dragging water from inside of the Gulf of Ana María with high nutrient content influenced by human activities towards the extremes of the Park, which allows this water to reach the lagoons and channels where *Cassiopea* individuals mostly live. These three zones are located at points where there is an important hydrological exchange between the Gulf of Ana María and the Caribbean Sea: Pasa de Caballones, Pasa de Boca Grande, and Boca Rica Channel (*González-de Zayas et al., 2006*). At these exchange points, nutrient-rich water from the land and the gulf flow through these passes. The proximity of these sites to these exchange zones could be a reasonable explanation for the high bell size values for the three zones. In the case of Zone 7, which presents the smallest sizes, it is due to the fact that most of the individuals sampled were juveniles that do not exceed 5 cm in bell diameter. This zone may be a nursery area or where greater recruitment of larvae is occurring, studies are required to address these issues more acutely from the geographical characteristics of this Zone 7.

The highest mean density reported for the eight zones corresponds to Zone 7, 2 and 5, which in turn were the zones with the lowest values of bell diameter. Then, it would be reasonable to infer that, with increasing size, density would decrease, taking into account that the number of individuals that could occupy 1 $m^2$ would decrease. A study in Barbuda also observed a negative relationship between bell diameter and density in *Cassiopea* populations (*Zarnoch et al., 2020*). On the other hand, it has also been suggested that the size of the *Cassiopea* bell may be related to the availability of nutrients, since in general jellyfish in conditions of low nutrient availability reduce their diameter, which may be an evolutionary adaptation to survive in low-nutrient environments. (*Stoner, Archer & Layman, 2022*). Also, the density of these jellyfish is known to be related to the distribution of *T. testudinum* present in the area (*Stoner, Yeager & Layman, 2014a*). The areas with the highest density of *Cassiopea* are in turn the areas with the lowest coverage of *T. testudinum* (Table 3). Moreover, the low density of individuals present in Zone 3 could be related to this site being completely devoid of red mangroves, which are known to provide key peptides for larval settlement and for the development of individuals (*Fleck & Fitt, 1999*). This coincides with the recent red mangrove mortality events reported by *Pina-Amargós, Figueredo-Martín & Rossi (2021)*.

When analyzing the differences in bell diameter (size) between the three main regions (East, Central, and West), although they were significantly smaller in the West Region, this can be explained by the high number of individuals smaller than five cm found in Zone 7. If we exclude the bell diameter values of Zone 7 in the analysis, in general, the bell diameter of the three regions do not show great differences, moving in a range between 12. 22–12.73 cm. This may mean that, there are no major ecological differences between the three regions of the JRNP (at least not for a significant effect on the size of the animals), where the sampled zones did not differ much in terms of substrate types, presence of red mangrove despite the reported mortality, and the import of nutrients by water exchange

from the mainland to the Caribbean Sea that allows *Cassiopea* individuals to reach similar average diameters in the three regions. Nutrient availability (natural or introduced by human activity), influences the growth of the *Cassiopea*, so it could be a factor that would further influence the relationship that exists between bell diameter and density.

In general, there was a greater presence of *C. xamachana* in the JRNP, with 7,618 individuals out of a total of 10,803. Although in some sites one species are dominant over the other, in general, they are found to share the same ecological niche. These results are unusual considering that these two species of *Cassiopea* are described at different depths and on different substrates. It is proposed that *C. xamachana* is found in muddy substrates (*e.g.*, mangroves) and at a maximum depth of 1.5 m while *C. frondosa* is found in a coarse sediment substrate (*e.g.*, sand and reef) and a depth range of 1 to 5 m (*Larson, 1997*). In this work, no marked differences were found in the presence or absence of both species in terms of depth, temperature, or substrate. Taking into account that the range in which *C. frondosa* is found is much wider than that of *C. xamachana*, in Peralta the number of *C. frondosa* individuals was greater, considering that the depth range was between 0.40 and 2.30 m.

*Fitt et al. (2021)* conducted a study to compare the physiology of *C. xamachana* and *C. frondosa* in Florida. The results showed that the presence of a high-temperature resistant symbiont in *C. xamachana* such as *Symbiodinium microadriaticum* influences the distribution of the species in shallow waters, while *C. frondosa* harbors a heat-sensitive symbiont such as *Breviolum* sp. (ITS-type B19) so it needs deeper habitats because these tend to be colder. Because of this, *Fitt et al. (2021)* posit that *C. xamachana* is displacing *C. frondosa* to shallow warm waters for populations sampled in Florida, USA. Considering that the temperature ranges between Florida and Cuba do not differ that much, it is possible that for the populations sampled in the JRNP, *C. xamachana* is also displacing *C. frondosa* to shallower and warmer waters. However, this niche segregation was not found in this study.

The percentage of benthic cover for *Cassiopea* is above the values found in the literature for the genus. The highest cover values for *Cassiopea* were found in Laguna de las Anclitas (23.72%) and Boca de las Anclitas (23.22%), both in Zone 4 of the JRNP, and Canal de Caballones (22.33%) in Zone 5. These two sites are where there is usually more tourist activity with boats for recreational fishing and diving, therefore a higher level of disturbance compared to the other sampled areas (*Hernández-Fernández et al., 2018*). These activities are known to contribute extra nutrients to the nearby waters, which benefits jellyfish of *Cassiopea* (*Stoner et al., 2011*), so this could explain the high cover values of individuals from these sites compared to the other sites sampled (Table S2).

The highest percentages of *T. testudinum* were found in Laguna de las Anclitas Noroeste (70.37%) in contrast to the cover value for *Cassiopea* of 10.98%, This could be due to the fact that jellyfish are known to have a negative effect on seagrass beds, so such *T. testudinum* cover values compared to *Cassiopea* benthic cover values evidence this (*Stoner, Yeager & Layman, 2014a*; *Stoner et al., 2014b*). *Cassiopea* has been found resting on top of seagrass blades, so it has been suggested that in areas where *Cassiopea* is abundant, seagrass cover is reduced (*Stoner et al., 2014b*). One of the reasons for reduced seagrass cover is that a high

abundance of *Cassiopea* individuals prevents sunlight from reaching the seagrass leaves, in turn inhibiting the gas exchange that occurs at the surface of their leaves (*Stoner et al., 2014b*). In addition, the nighttime respiration of *Cassiopea* reduces dissolved oxygen concentrations (*Verde & McCloskey, 1998*), limiting seagrass development. Finally, bell pulsations generate a flow that creates disturbances in seagrass shoots, reducing shoot stability in the sediment (*Stoner et al., 2014b*).

On the contrary, some authors suggest that the presence of *Cassiopea* could somehow benefit seagrasses, considering that the mucus they release contains cnidocytes that are harmful to some species of herbivorous fish (*Niggl et al., 2010*; *Stoner et al., 2014b*). In the case of this study, only two individuals of the Urotrygonidae family and a single individual of the Sphyraenidae family were found at and/or near the sampling sites. A few specimens of *Lutjanus apodus* were also observed in areas very close to the mangrove. In a study conducted in the Bahamas by *Stoner et al. (2014b)* they found that *Gerres cinereus* dies just after swimming through the water column containing mucus released by *Cassiopea*, so the presence of the jellyfish could limit the movements of fauna associated with seagrass beds and mangroves, such as fish.

The invertebrate cover was found to a lesser extent, which is evidence that *Cassiopea* competes for space in the seagrass beds with some invertebrates. *Stoner, Yeager & Layman (2014a)* found that high densities of *Cassiopea* may limit the space available for other sessile organisms.

## CONCLUSIONS

The presence of *Cassiopea* spp. was detected in all the zones sampled in the JRNP, presenting an aggregated spatial distribution. The highest density was found in Zone 7, while Zone 3 was the site with the lowest density. In general, the density values for the JRNP coincide with those reported for the Caribbean region. A greater size of individuals was observed in Zone 4 and this may be related to a greater anthropization of the area and greater aggregations of the species were observed at smaller sizes. *C. xamachana* and *C. frondosa* share a niche and no evident segregation was detected according to abiotic variables. The percentage cover of *Cassiopea* was higher than those reported in the literature. The bell diameter sizes of *Cassiopea* spp. populations in the JRNP did not differ because the habitat characteristics were similar. The Eastern Region showed differences in density from the other regions. Despite the differences in density, habitat characteristics at the fourteen sites sampled were similar, so there are no ecological differences in *Cassiopea* populations in the three regions of the JRNP. More studies would be needed on the Cuban coast to describe the possible relationship between bell diameter and density of *Cassiopea* spp. as well as its importance and possible applications.

## ACKNOWLEDGEMENTS

The authors express their gratitude to the crew of the research vessel "Ocean for Youth" for their assistance in the expeditions. To all the organizations that made the expeditions to the Jardines de la Reina National Park possible: The Ocean Foundation, The Ocean for Youth,

Sweet-Avalon, International Chair of Conservation and Management of Marine-Coastal Ecosystems of the Harte Research Institute of the University of Corpus Cristi Texas A&M, and the Environmental Defense Funds. Thanks to Dr. Fabián Pina Amargós and Dra. Patricia González Díaz for their advice and suggestions. Thanks to Laura del Río, Ariandy González, Eddy García, and colleagues from the CIM-UH for their technical assistance. We greatly thank our editor (Christopher Glasby) and reviewers for their valuable comments and suggestions on our MS.

### Funding

The authors received no funding for this work.

### Competing Interests

The authors have declared that no competing interests exist.

### Author Contributions

- Ramón Damián Morejón-Arrojo performed the experiments, analyzed the data, prepared figures and/or tables, authored or reviewed drafts of the article, and approved the final draft.
- Leandro Rodriguez-Viera conceived and designed the experiments, performed the experiments, analyzed the data, prepared figures and/or tables, authored or reviewed drafts of the article, and approved the final draft.

### Field Study Permissions

The following information was supplied relating to field study approvals (*i.e.*, approving body and any reference numbers):

Field experiments were approved by Ministerio de Ciencia Tecnología y Medio Ambiente. Oficina de Regulación de Seguridad Ambiental. No. 5 de 2021.

### Data Availability

The raw data is available in the Supplemental File.

### Supplemental Information

Supplemental information for this article can be found online at http://dx.doi.org/10.7717/peerj.15254#supplemental-information.

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
