# Peer review of "Characterization of the populations of upside-down jellyfish in Jardines de la Reina National Park, Cuba"

_PeerJ, doi:10.7717/peerj.15254_

## Round 0.1 · original submission · Major Revisions

· Academic Editor

Major Revisions

Dear Author: the reviewers have provided detailed and constructive reviews which include both compliments and recommendations for improvement. To summarize the latter, they suggest improvements in the type of information presented and the organization of the information. Type of information - for example, more details on the experimental design, environmental measurements that were undertaken, and identification of the jellyfish, including differences between the two species and how the young animals were identified (or not), clarification of seasonal variation vs habitat. Organization of information, for example, one reviewer suggests reorganizing and reducing the information from the material and methods section, trying to avoid redundancy in the discussion section, and reviewing the necessity of all figures and tables. Please note that the journal provides a supplementary information section - consider moving any material that is non-essential here.

I hope you find the reviewer's appraisals useful - in considering them, please focus on making the manuscript more informative and more concise.

Thank you for your submission and I look forward to seeing your revised version, best Chris

Reviewer 1 ·

Basic reporting

This is a very interesting and relevant manuscript detailing the presence of Cassiopea in JRNP. Overall, the manuscript is well structured and written, and includes relevant literature, figures and tables. However, I do have a few minor comments below that need to be addressed:

Line 23: the authors state that this study is significant because no other studies have analysed the densities of Cassiopea in seagrass or mangrove ecosystems. However, this is not the case. The authors need to revise this angle and read Stoner et al. (2011) Effects of anthropogenic disturbance on the abundance and size of epibenthic jellyfish Cassiopea sp.. The authors can revise this sentence to say that there have been no studies of the densities of Cassiopea in seagrass or mangrove systems in Cuba.
Line 35: confused about the reporting of the stats. If 2.18 to 14 is the range, what is the 52 and the 79 representing? Do the authors mean range and interquartile range? This needs to be clarified throughout the manuscript
Line 37: Similar to line 35.
Line 64: Cassiopea is spelt incorrectly and needs to be in italics. Please check throughout
Line 138: I suggest rephrasing “for compare” to “to compare”
Line 174: When the authors present the mean in the results section, are they presenting the standard error as well? This needs to be reported as such, for example “mean (± SE)”.
Line 184: Same as comment for line 35
Line 188: I think it would be good to state the depth range that Cassiopea occurred in as they generally prefer shallow areas, and for some sites may not occur at 4m depth. Did the quadrats placed down cover this full depth range?
Lien 207: is this the mean and standard error of the density?
Line 207: This is up to the author, but I would rephrase cassiopeas to Cassiopea in italics.
Line 213: Is this presentation of the mean and standard error?
Line 243: is 32 PSU the mean? Additionally, if possible I would split this sentence into two.
Line 252: “salinity is not a factor” would be a better way of phrasing this
Line 254: Temperature presented in December?
Line 262: seasonal variations in rain? Also I suggest rephrasing as “ … and future studies…”
Line 264: what is meant by development? Population growth?
Line 281: I would suggest replacing “We could think…” to “This zone may be a nursery area…” or something similar
Line 290: Cassiopea needs to be in italics
Line 292: Same suggestion as line 207. Additionally, I suggest removing “that” in “since that jellyfish”
Line 296: same as line 207
Line 303: Is length the same as diameter? Make sure terminology is consistent throughout the manuscript
Line 305: What values are being excluded? The diameters?
Line 307: what is the value of the 12 before the range? Is this the mean? Additionally, the range should be presented lowest to highest
Lien 313: remove “jellyfish”
Line 354: Cassiopea needs to be in italics
Line 367: reference needed
Line 380: something is missing before “mainly fish”, I suggest “such as fish” or something similar.
Line 392: It would be good to name the two species found in your conclusion
Figure 1: It would be good to have the orange squares showing activity also in the legend of the figure
Figures 2,3, 4, 5,6: Need to state what the error bars are. Are they standard error?
Figure 3: The x axis title should be ‘zone’
Figure 7: Main title "data 1" needs to be removed from the figure
Table 5: Is this mean and standard error?

Experimental design

The authors have presented original primary research that is able to accurately report on the presence of Cassiopea within the national park, which has not been reported on before. As a result, the authors were able to fill in a knowledge gap with detail. However, I think a bit more information is required for how and where the quadrats were randomly placed. Was a random number generator used? At what depths were the quadrats placed? Cassiopea densities can vary a lot with depth, and so I think it would be good if the authors state the depth of the quadrats as well as the sites. Additionally, Cassiopea is considered to be a cryptic genus and distinguishing species can be difficult. The authors state they were able to identify two species of Cassiopea in the area, but I think more details is needed in regards to this. Did the authors collect any voucher specimens? Were the species identified from photos or in the field? Additionally, turbidity can impact the presence of Cassiopea, did the authors take any turbidity measurements? If not, this should be mentioned in the discussion.

Validity of the findings

The authors have done a great job replicating the design and presenting the data. The rational and benefit of the experiment is clearly stated, and the results are statistically sound. Additionally, the conclusion and takeaway message are clear.

Reviewer 2 ·

Basic reporting

no comment

Experimental design

no comment

Validity of the findings

no comment

Additional comments

General comments:

1. In some parts of the manuscript, phrasing makes comprehension difficult. I suggest you have a colleague who is proficient in English and familiar with the subject matter review your manuscript one more time.
2. Reorganize and reduce the information from the material and methods section.
3. It is common to read that morphological variations make it difficult to identify Cassiopea species. Can you mention the differences in shape, size, and color that you used to distinguish both species? A picture of each species would be useful to show their morphological differences.
4. Was it possible to identify young medusae at the species level?
5. Try to avoid redundancy in the discussion section. For example, by omitting results that were already mentioned in the results, figures, and tables.
6. I know that some sampling sites were visited in December, others in February-March and others in July-August. Could differences in size and density be related to seasonal variations instead of habitat characteristics? Does the presence of juveniles in zone 7 could be related to the breeding season of these animals?
7. The number of figures and tables can be reduced.

Specific comments:

Line 26 – Avoid acronyms that are not used multiple times in the abstract (MPA).
Line 30-31 – The sentence “A total of 10 803 individuals were recorded, of which 7618 Cassiopea xamachana and 3185 C. frondosa.” seems to be incomplete or needs to be rewritten to make it understandable.
Line 40 – The entire genus name must be spelled out on first use.
Lines 53-56 – The current phrasing makes comprehension difficult. In my opinion, this sentence could be transformed into two, one of them talking about the species richness of Cassiopea and the other one about its distribution/habitat.
Line 58 – Make double check to confirm that Odhera et al., mentioned that the epibenthic lifestyle is an adaptation to symbiosis.
Line 61 – In all the paragraphs Cassiopea has been used as a plural. So, I suggest using “they need” instead of “it needs”.
Line 64 – Cassiopea instead of Cassiopeia.
Line 71-75 – Assuming that these lines are a paragraph, it is necessary to add supporting sentences to develop the unique sentence/idea presented.
Line 76 – This paragraph starts in the same way that the previous one (“This genus of jellyfish”).
Line 76 – The information about the habitat/distribution of Cassiopea from lines 55-56 (“inhabits shallow tropical and subtropical waters such as mangroves and seagrass beds”) could be incorporated here.
Line 79 – “the genus Cassiopea”, the word “genus” can be deleted since you already said that Cassiopea is a genus, check it along the document.
Line 91 – What kind of lists? Species lists?
Line 93 – “However, these are limited.” Do you refer to the ecological studies or the studies of these jellyfish in Cuba?
Line 103 – “three keys”, do you mean three "big" keys or regions?
Study area section – In my opinion, some information could be omitted. E.g., “Given the importance of the park over the years it has received numerous specialists from different disciplines, both marine and terrestrial.”, “This gradient has also been detected for coral species in the JRNP (Hernández-Fernández et al., 2018).”, etc.
Line 116 – Which sites were included in each zone?
Line 136 – “C. frondosa” the entire genus name must be spelled out on first use.
Line 137 – What are the differences in shape, size, and color of appendages between the two species?
Line 140 – “25 %” in English, the symbol for a percentage follows the number without any space between.
Line 141 – “spp.,” instead of “spp,”
Line 141 – “S. filiforme”, the entire genus name must be spelled out on first use.
Line 147-151 – The zones should be specified on the first mention (line 116).
Line 152-155 – The regions could also be specified in the study area section.
Line 176 – In line 30, the authors used space as a thousand separator (10 803) and in this line, they used a comma (10,803), choose one format and homogenize the document.
Line 176-181 – All of these results appear in table 4, so it is not necessary to write all of them once again.
Line 189 – It seems that the degree sign is not the correct one “0C” -> °, check it along the document.
Line 191 – Is there a relationship between the bell size and the sampling season?
Line 192 – Is PNJR the same that JRNP?
Lines 200-204 – Those results also appear in figure 3 and Table 5, try to avoid redundancy.
Line 205 -- Is there a relationship between the density and the sampling season?
Line 222 – It is common to read that morphological variations make it difficult to identify Cassiopea species, could you provide a picture of the two species?
Line 235 – Corals instead of “coral´s”
Line 238 – The entire genus name must be spelled out if it begins a sentence.
Line 243 – In line 189, the authors said the salinity “ranged from 32.25 to 37.32” and here they say that it ranged from 34.5 to 37.32, which one is correct?
Line 243 – The authors used a period as a decimal separator (34.5) and a comma (37,32); homogenize the format along the document.
Lines 245 and 247 – “X %.”, in English, the symbol for a percentage follows the number without any space between, check it along the document.
Line 250 – I suggest using “recorded” instead of “obtained”.
Line 254 – “26.8” define the number of decimals that are going to be used (in line 184 the authors used 2).
Line 255 – Since Aljbour worked with an unidentified species, I suggest saying “jellyfish” instead of “species”.
Line 256-258 – the current phrasing makes comprehension difficult.
Line 257 – As far as I understand, not all the samplings were performed in December, right? Did you observe a relationship between abundance and sampling month?
Line 262 – You collected specimens in December, Feb-March and July-August, right? Did you observe seasonal variations regarding density?
Lines 284-287 – These are results and they already appeared in the results section.
Line 289 – Cassiopea should be in italics.
Line 292: is “cassiopeas” another common name for Cassiopea?
Line 293 – Stoner, Archer & Layman (2022) mentioned “…may be an evolutionary adaption”
Line 303 – I suggest changing “lengths” to “bell diameter (size)”
Line 307-308 – “This may mean that, there are no major ecological differences between the three regions of the JRNP (at least not for a significant effect on the size of the animals) …” So, there is no difference regarding the size of the animals and the sampling season?
Lines 318-326 – This information appears in table 4, try to avoid redundancy.
Line 339 – Shallow water instead of “shallow and shallow waters”?
Line 346 – At least four paragraphs end saying that more studies are needed on the Cuban coast (lines 262, 282, 315, 346). Try to avoid redundancy by including this statement in the conclusion section.
Line 354 – Cassiopea in italics. The word “genus” can be deleted.
Lines 358-360 – In my opinion, the information “… in Cayo Piedra Piloto 39.66 % and for Cassiopea 13.79 %, Boca de las Anclitas Este 35.20 % and 14, 84 % for Cassiopea and Cachiboca 27.13 % and 6.84 % for Cassiopea.” can be deleted. That is because the information provided by Laguna de las Anclitas Noroeste is enough to support the idea that you develop in the next sentence.
Lines 376 and 377 – The authors wrote the taxonomic authorities for the species Lutjanus apodus and Gerres cinereus, but they did not for C. frondosa (neither for other invertebrates nor plants). Decide for which taxa you are going to mention the taxonomic authorities.

Figures and tables:

Figure 1. I cannot see (in the figure) the legends “REE: Reserve Extreme East, REW: Reserve Extreme West.” (They were mentioned in the figure legend). I suggest coloring the sampling sites by zone.
Figure 3. In the X-axis label use zone instead of “zonaçe”. In the Y-axis labels use lowercase for “density”.
Figures 2 and 3 are similar (they have the same scale in the X-axis and a similar one on the Y-axis). They could be mashed up into only one figure.
In my opinion, figures 4 and 5 are not essential, those results can be simply mentioned in the text and the figures can be included in the supplementary material. In both cases, replace “O” with “W” in the X-axis legend or specify that “O” means west.

Tables. The authors used commas as decimal separators in the tables but along the manuscript, they also used peridots (e.g., line 260); choose one and homogenize the format.
In my opinion, there are a lot of tables in the manuscript. Even when there are no print restrictions, for readability purposes, I encourage authors to include no more than 3-4 tables. The other ones could be included as supplementary appendices.

---

## Round 0.2 · Minor Revisions

· Academic Editor

Minor Revisions

Dear Author

Thank you for the revised manuscript - it is looking very nice. One reviewer has had a second look over the manuscript and identified a couple of further minor checks/additions. Also, in the attached pdf, I provide my editorial comments, for incorporation into your final version. Please note that you should also:
1. provide keywords
2. cite the supplementary files in the text
3. Provide a short caption for each of the supplementary files at the end of the manuscript.

Best Chris

Reviewer 1 ·

Basic reporting

The authors have done a great job addressing all of the comments and revising the manuscript. However, I do have two minor comments that need to be addressed before the paper is published.

1. Line 147: References need to be included to support the authors comment about the differing number of oral arms and branching pattern between the two species. This will make the authors distinction more robust.
2. Check spelling of Cassiopea and ensure that it is in italics throughout the manuscript (e.g. Line 211, 236, 245).

Experimental design

No comment

Validity of the findings

No comment

---

## Round 0.3 · accepted · Accept

· Academic Editor

Accept

Dear Author, thanks for addressing those final issues identified by myself and one of the original reviewers. I am happy with the current version and will recommend that it now go to production. All the best Chris